# Online dynamical learning and sequence memory with neuromorphic nanowire networks

Ruomin Zhu [1,9] ✉, Sam Lilak[2,9], Alon Loeffler[1], Joseph Lizier [3,4], Adam Stieg [5,6] ✉, James Gimzewski[2,5,6,7] ✉ & Zdenka Kuncic [1,4,8] ✉

Nanowire Networks (NWNs) belong to an emerging class of neuromorphic systems that exploit the unique physical properties of nanostructured materials. In addition to their neural network-like physical structure, NWNs also exhibit resistive memory switching in response to electrical inputs due to synapse-like changes in conductance at nanowire-nanowire cross-point junctions. Previous studies have demonstrated how the neuromorphic dynamics generated by NWNs can be harnessed for temporal learning tasks. This study extends these findings further by demonstrating online learning from spatio-temporal dynamical features using image classification and sequence memory recall tasks implemented on an NWN device. Applied to the MNIST handwritten digit classification task, online dynamical learning with the NWN device achieves an overall accuracy of 93.4%. Additionally, we find a correlation between the classification accuracy of individual digit classes and mutual information. The sequence memory task reveals how memory patterns embedded in the dynamical features enable online learning and recall of a spatiotemporal sequence pattern. Overall, these results provide proof-of-concept of online learning from spatiotemporal dynamics using NWNs and further elucidate how memory can enhance learning.

Neuromorphic devices offer the potential for a fundamentally new computing paradigm, one based on a brain-inspired architecture that promises enormous efficiency gains over conventional computing architectures[1–11]. A particularly successful neuromorphic computing approach is the implementation of spike-based neural network algorithms in CMOS-based neuromorphic hardware[2,12–17]. An alternate neuromorphic computing approach is to exploit brain-like physical properties exhibited by novel nano-scale materials and structures[18–22], including, in particular, the synapse-like dynamics of resistive memory (memristive) switching[4,23–31].

This study focuses on a class of neuromorphic devices based on memristive nanowire networks (NWNs)[32,33]. NWNs are comprised of metal-based nanowires that form a heterogeneous network structure similar to a biological neural network[34–38]. Additionally, nanowire-nanowire cross-point junctions exhibit memristive switching attributed to the evolution of a metallic nano-filament due to electro-chemical metallisation[39–43]. Typically, each NWN contains thousands of nanowires and an even greater number of junctions. In response to electrical input signals, NWNs also exhibit brain-like collective dynamics (e.g., phase transitions, switch synchronisation, avalanche

[1]School of Physics, The University of Sydney, Sydney, NSW, Australia. [2]Department of Chemistry and Biochemistry, University of California, Los Angeles, Los Angeles, CA, US. [3]School of Computer Science, The University of Sydney, Sydney, NSW, Australia. [4]Centre for Complex Systems, The University of Sydney, Sydney, NSW, Australia. [5]California NanoSystems Institute, University of California, Los Angeles, Los Angeles, CA, US. [6]WPI Center for Materials Nanoarchitectonics (MANA), National Institute for Materials Science (NIMS), Tsukuba, Japan. [7]Research Center for Neuromorphic AI Hardware, Kyutech, Kitakyushu, Japan. [8]The University of Sydney Nano Institute, Sydney, NSW, Australia. [9]These authors contributed equally: Ruomin Zhu, Sam Lilak. ✉e-mail: rzhu0837@sydney.edu.au; stieg@cnsi.ucla.edu; gim@chem.ucla.edu; zdenka.kuncic@sydney.edu.au

criticality), resulting from the interplay between memristive switching and their recurrent network structure[34,37,44–50].

Recurrent, sparse networks can transform temporal signals into a higher-dimensional dynamical feature space[51,52], which is advantageous for machine learning applications involving dynamically evolving data[53]. Furthermore, the computational burden of training network weights can be circumvented altogether by leveraging Reservoir Computing (RC), which restricts training to a linear output layer, in which only linear weights need to be learned using the rich dynamical features generated by the recurrent network reservoir[54–56]. Physical systems are particularly useful as reservoirs, due to their self-regulating dynamics and physical constraints imposed by conservation laws (e.g., Kirchoff's laws), in contrast to algorithmic RC, which typically uses a random network with fixed weights and requires manual hyper-parameter optimisation[57]. Previous experimental[36,58,59] and simulation[36,58,60–65] studies have demonstrated NWNs exhibit fading memory and can effectively project input signals to a higher-dimensional feature space, thus enabling their use as physical reservoirs in an RC approach to machine learning.

In previous physical RC studies, learning is achieved by training the readout weights after the entire input stream is delivered to the physical system[66], while the real-time response from the network is not fully reflected in the learning outcome. While such batch-based approaches can be practically limited by memory availability when working with large datasets, an arguably more important consideration is the need to re-train weights when feature distributions evolve[67]. An alternate approach is online training, which has the potential to enhance dynamical learning by allowing the readout weights to adapt to non-stationary dynamical features incrementally[68,69]. As is the case

for conventional machine learning, online learning approaches are necessary for scaling up neuromorphic computing and ultimately achieving the goal of continual learning[70–72].

In this study, we use an NWN device to demonstrate *online dynamical learning*, i.e., learning incrementally from continuous streams of dynamical features. We implement an online training algorithm within an RC framework and use the MNIST handwritten digit database to deliver a stream of spatiotemporal patterns to the NWN device. Dynamical features in the device readouts are then used to train a linear classifier in an online manner, sample by sample, and information-theoretic measures are used to analyse the online learning process. By constructing a numerical sequence pattern using the MNIST database, we then develop and implement a novel sequence memory task that demonstrates the NWN's ability to generate spatiotemporal memory patterns in a similar manner to the brain, using attractor dynamics. We show how these sequence memory patterns can also be learned in an online manner and then used to recall a target digit presented earlier in the sequence. By comparing recall performance with and without memory patterns, we demonstrate how memory enhances learning.

## Results

The first task we performed to test online dynamical learning is the MNIST handwritten digit classification task, which has not previously been experimentally implemented on an NWN device (but has been implemented in NWN simulations[62,65]). A schematic illustration of the experimental setup for the online classification of MNIST handwritten digits using an NWN multi-electrode array (MEA) device is shown in Fig. 1. MNIST digit images[73] are converted to 1-D temporal voltage

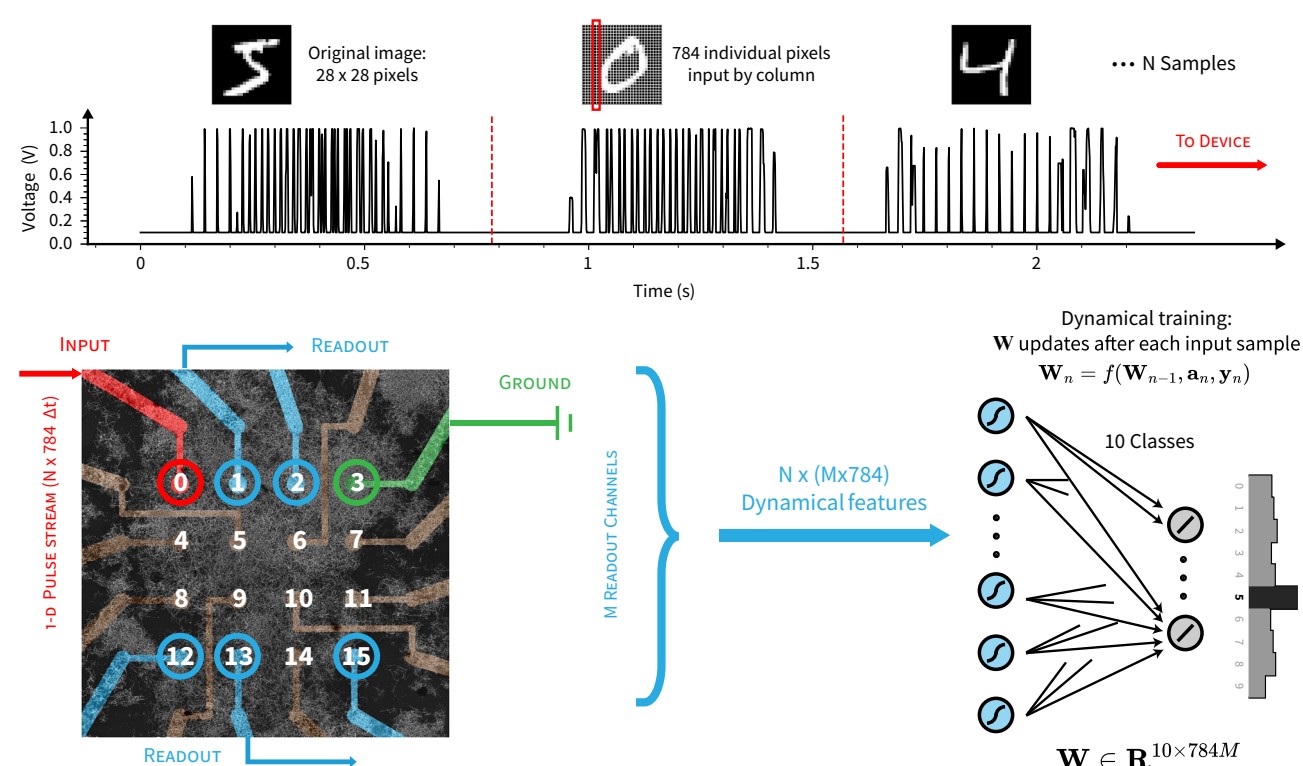

**Fig. 1 | Schematic illustration of nanowire network device setup for MNIST digit classification demonstrating online dynamical learning.** Top: MNIST handwritten digit samples ($N$ samples × 784 pixel features) are normalised and converted to 1-D temporal voltage pulse streams (each pixel occupies $\Delta t = 0.001\,s$) and delivered consecutively to the nanowire multi-electrode device. Bottom left: scanning electron micrograph image of the 16-electrode device, showing source electrode (channel 0, red), drain electrode (channel 3, green), readout electrodes (channel 1, 2, 12, 13, 15, blue) and other electrodes not used (brown). Bottom right: readout voltages (i.e., $N \times M \times 784$ dynamical features) are input into an external linear classifier in which the weight matrix $\mathbf{W}_n$ for the $M \times 784$ features per digit sample is updated after each sample $\mathbf{a}_n$, with corresponding class $\mathbf{y}_n$ as the target output (digit '5' shown as an example of classification result).

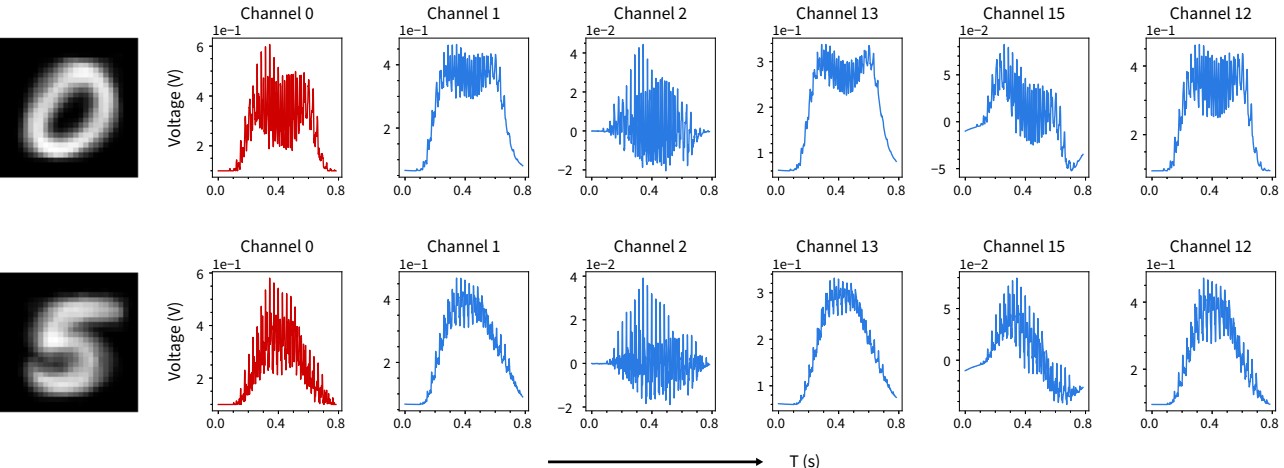

**Fig. 2 | Input–output mapping of MNIST digits in a nanowire network device.** Images of MNIST digits '0' and '5' averaged across 100 samples randomly selected from the training set (column 1), their corresponding input voltage streams (column 2, red) and readout voltages from multiple channels (columns 3–7, blue).

pulse streams and delivered consecutively to one electrode. The network's real-time response is read out from other electrode channels and classification is performed in an external (digital) fully-connected output layer. The weights are learned from the dynamical features and updated after each digit sample using an online iterative algorithm based on recursive least squares (RLS). See "Methods" for full details.

### Dynamical feature generation

Figure 2 shows examples of handwritten digit image samples converted to 1-D voltage pulse streams delivered to the allocated source electrode (channel 0) and the corresponding voltage streams read out from other channels (1, 2, 12, 13 and 15) for the setup shown in Fig. 1 (readouts for other channels and digits are shown in Supplementary Figs. S4 and S5).

Each row in Fig. 2 shows the averaged image, input and readout data for 100 MNIST samples randomly selected from the training set for the corresponding digit class.

For each class, the readout voltages from each channel (columns 3–7, blue) are distinctly different from the corresponding input voltages and exhibit diverse characteristics across the readout channels. This demonstrates that the NWN nonlinearly maps the input signals into a higher-dimensional space. Rich and diverse dynamical features are embedded into the channel readouts from the spatially distributed electrodes, which are in contact with different parts of the network (see Supplementary Fig. S6 for additional non-equilibrium dynamics under non-adiabatic conditions). We show below how the inter-class distinctiveness of these dynamical features, as well as their intra-class diversity, can be harnessed to perform online classification of the MNIST digits.

**Table 1 | MNIST digit classification using an NWN device with one and five readout channels for online and batch (offline) linear classifiers**

| Readout data | Classifier | Accuracy |
|---|---|---|
| 784 × 1 | Online | 91.6% |
| 784 × 5 | Online | 93.4% |
| 784 × 1 | Batch | 86.6% |
| 784 × 5 | Batch | 91.4% |

Each classification accuracy is calculated for 10,000 testing samples and averaged across 5 iterations. The online classifier is trained for a single epoch of 50,000 training samples, while the batch-based classifier is trained and optimised using gradient descent for 100 epochs with 500 mini-batches of size 100.

### Online learning

Table 1 presents the MNIST handwritten digit classification results using the online method (external weights trained by an RLS algorithm). Results are shown for one and five readout channels. For comparison, also shown are the corresponding classification results using the offline batch method (external weights trained by backpropagation with gradient descent). Both classifiers learn from the dynamical features extracted from the NWN, with readouts delivered to the two classifiers separately. For both classifiers, accuracies increase with the number of readout channels, demonstrating the nonlinearity embedded by the network in the readout data. For the same number of channels, however, the online method outperforms the batch method. In addition to achieving a higher classification accuracy, the online classifier $W$ requires only a single epoch of 50,000 training samples, compared to 100 training epochs for the batch method using 500 mini-batches of 100 samples and a learning rate $\eta = 0.1$. The accuracy of the online classifier becomes comparable to that of the batch classifier when active error correction is not used in the RLS algorithm (see Supplementary Table 1). A key advantage of the online method is that continuous learning from the streaming input data enables relatively rapid convergence, as shown next.

To better understand how learning is achieved with the NWN device, we investigated further the dependence of classification accuracy on the number of training samples and the device readouts. Figure 3a shows classification accuracy as a function of the number of digits presented to the classifier during training (See Supplementary Fig. S7 for classification results using different electrode combinations for input/drain/readouts and different voltage ranges). The classification accuracy consistently increases as more readout samples are presented to the classifier to update **W** and plateaus at ≃92% after ≃10,000 samples. Classification accuracy also increases with the number of readout channels, corresponding to an increase in the number of dynamical features (i.e., 5 × 784 features per digit for 5 channel readouts, the channels are added following the order 1,2,13,15,12) that become sufficiently distinguishable to improve classification. However, as shown in Fig. 3b, this increase is not linear, with the largest improvements observed from 1 to 2 channels. Figure 3c shows the confusion matrix for the classification result using 5 readout channels after learning from 50,000 digit samples. The classification results for 8 digits lie within $1.5\sigma$ (where s.d. is $\sigma = 3\%$) from the average (93.4%). Digit '1' demonstrates significantly higher accuracy since it has a simpler structure, and '5' is an outlier because of the irregular variances of handwriting and low pixel resolution (See Supplementary Fig. S8 for examples of misclassified digits).

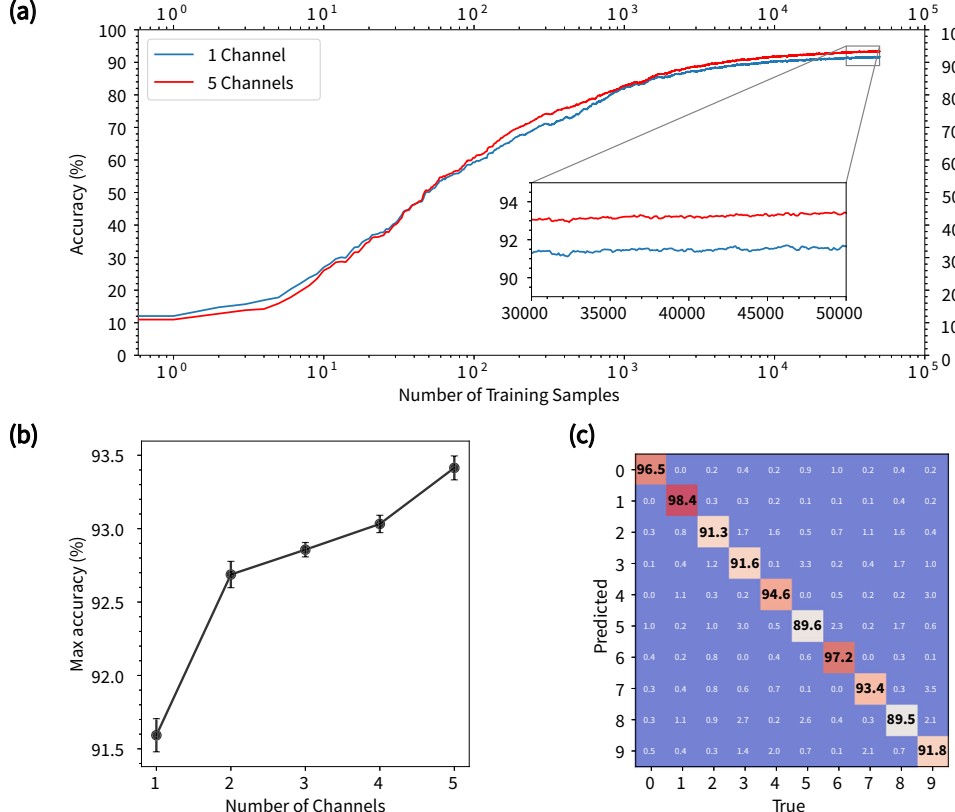

**Fig. 3 | Online learning of MNIST digits. a** Testing accuracy as a function of the number of training samples read out from one and five channels. Inset shows a zoom-in of the converged region of the curve. **b** Maximum testing accuracy achieved after 50,000 training samples with respect to the number of readout channels used by the online linear classifier. Error bars indicate the standard error of the mean of 5 measurements with shuffled training samples. **c** Confusion matrix for online classification using 5 readout channels.

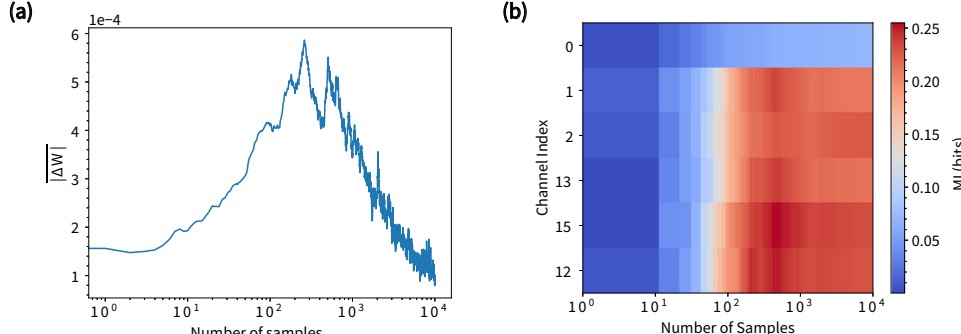

**Fig. 4 | Learning rate and mutual information. a** Mean of the magnitude of changes in the linear weight matrix, $\overline{|\Delta \mathbf{W}|}$, as a function of the number of samples learned by the network. **b** Corresponding Mutual Information (MI) for each of the 5 channels used for online classification (cf. Fig. 3) and for input channel 0.

## Mutual information

Mutual information (MI) is an information-theoretic metric that can help uncover the inherent information content within a system and provide a means to assess learning progress during training. Figure 4a shows the learning curve of the classifier, represented by the mean of the magnitude of the change in the weight matrix, $\overline{|\Delta \mathbf{W}|}$, as a function of the number of sample readouts for 5 channels. Learning peaks at $\simeq 10^2 - 10^3$ samples, after which it declines rapidly and becomes negligible by $10^4$ samples. This is reflected in the online classification accuracy (cf. Fig. 3a), which begins to saturate by ~$10^4$ samples. The rise and fall of the learning rate profile can be interpreted in terms of maximal dynamical information being extracted by the network. This is indicated by Fig. 4b, which presents mutual information (MI)

between the 10 MNIST digit classes and each of the NWN device readouts used for online classification (cf. Fig. 3). The MI values for each channel are calculated by averaging the values across the 784 pixel positions. The coincidence of the saturation in MI with the peak in $\overline{|\Delta \mathbf{W}|}$ between $10^2 - 10^3$ samples demonstrates learning is associated with information dynamics. Note that by $\simeq 10^2$ samples, the network has received approximately 10 samples for each digit class (on average). It is also noteworthy that MI for the input channel is substantially smaller.

Figure 5 shows MI estimated in a static way, combining all the samples after the whole training dataset is presented to the network. The MI maps are arranged according to the digit classes and averaged within each class. The maps suggest that distinctive information

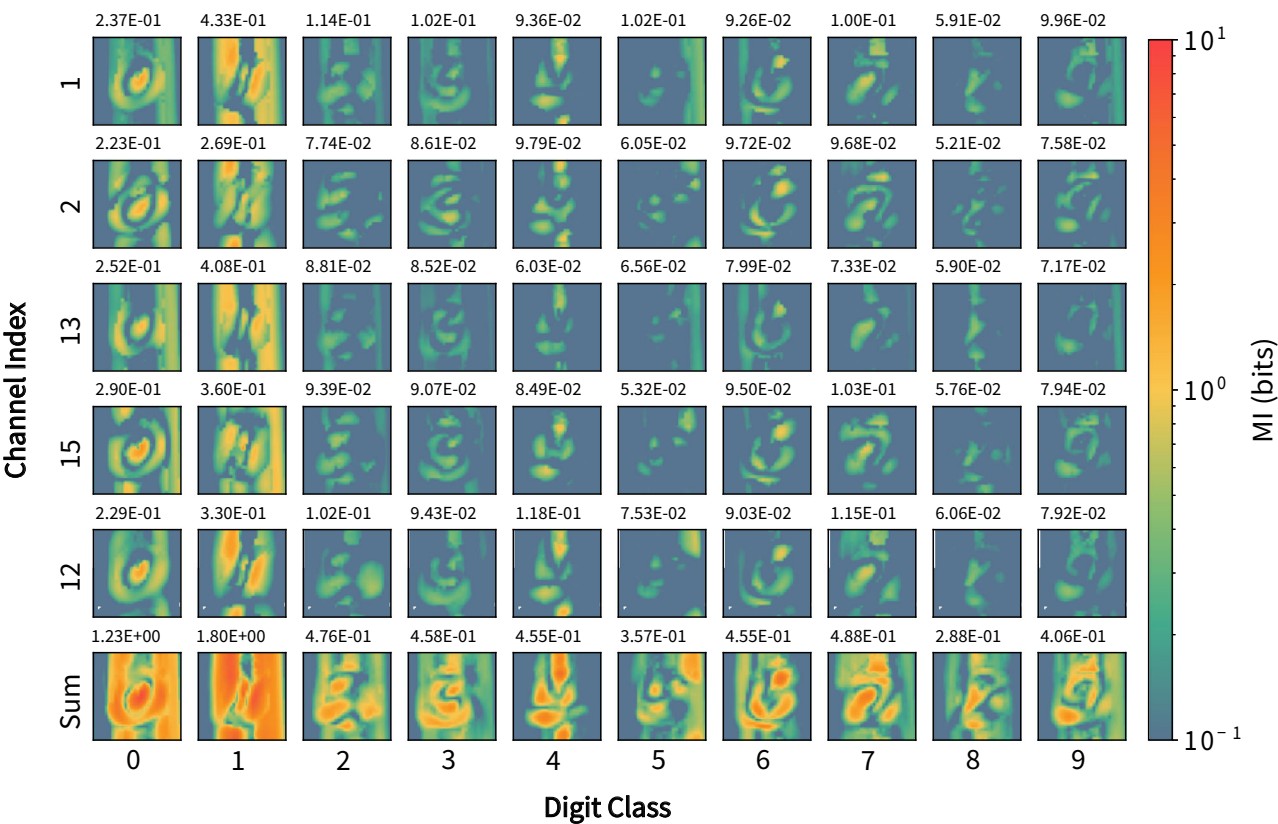

**Fig. 5 | Spatial distribution of mutual information (MI) for readout channels (rows), sorted by digit class (columns).** MI maps summed over all channels are shown in the bottom row, and mean MI is shown above each map.

content is extracted when digit samples from different classes are streamed into the network. This is particularly evident when comparing the summed maps for each of the digits (bottom row of Fig. 5). Additionally, comparison with the classification confusion matrix shown in Fig. 3c reveals that the class with the highest total MI value ('1') exhibits the highest classification accuracy (98.4%), while the lowest MI classes ('5' and '8') exhibit the lowest accuracies (89.6% and 89.5%), although the trend is less evident for intermediate MI values.

### Sequence memory task

As mentioned earlier, RC is most suitable for time-dependent information processing. Here, an RC framework with online learning is used to demonstrate the capacity of NWNs to recall a target digit in a temporal digit sequence constructed from the MNIST database. The sequence memory task is summarised in Fig. 6. A semi-repetitive sequence of 8 handwritten digits is delivered consecutively into the network in the same way as individual digits were delivered for the MNIST classification task. In addition to readout voltages, the network conductance is calculated from the output current. Using a sliding memory window, the earliest (first) digit is reconstructed from the memory features embedded in the conductance readout of subsequent digits. Figure 6 shows digit '7' reconstructed using the readout features from the network corresponding to the following 3 digits, '5', '1' and '4'. See "Methods" for details.

Figure 7a shows the network conductance time series and readout voltages for one of the digit sequence samples. The readout voltages exhibit near-instantaneous responses to high pixel intensity inputs, with dynamic ranges that vary distinctively among different channels. The conductance time series also exhibits a large dynamic range (at least 2 orders of magnitude) and, additionally, delayed dynamics. This can be attributed to recurrent loops (i.e., delay lines) in the NWN and to memristive dynamics determined by nano-scale electro-ionic

transport. The delay dynamics demonstrate that NWNs retain the memory of previous inputs (see Supplementary Fig. S9 for an example showing the fading memory property of the NWN reservoir). Figure 7b shows the respective digit images and $I-V$ curves for the sequence sample. The NWN is driven to different internal states as different digits are delivered to the network in sequence. While the dynamics corresponding to digits from the same class show some similar characteristics in the $I-V$ phase space (e.g., digit '1'), generally, they exhibit distinctive characteristics due to their sequence position. For example, the first instance of '4' exhibits dynamics that explore more of the phase space than the second instance of '4'. This may be attributed to differences in the embedded memory patterns, with the first '4' being preceded by '91' while the second '4' is preceded by '51' and both '9' and '5' have distinctively different phase space characteristics, which are also influenced by their sequence position as well as their uniqueness.

Figure 8a shows the image reconstruction quality for each digit in the sequence as a function of memory window length. Structural similarity (SSIM) is calculated using a testing group of 500 sets, and the maximum values achieved after learning from 7000 training sets are presented (see Supplementary Fig. S10 for the learning curving for $L = 4$ and Supplementary Fig. S11 for average SSIM across all digits). The best reconstruction results are achieved for digits '1' and '7', which are repeated digits with relatively simple structures. In contrast, digit '4', which is also repeated, but has a less simple structure, is reconstructed less faithfully. This indicates that the repeat digits produce memory traces that are not completely forgotten before each repetition (i.e., nano-filaments in memristive junctions do not completely decay). On average, the linear reconstructor is able to recall these digits better than the non-repeat digits. For the non-repeat digits ('5' and '9'), the reconstruction results are more interesting: digit '5' is consistently reconstructed with the lowest SSIM, which correlates with its low classification accuracy (cf. Fig. 3c), while '9' exhibits a distinctive

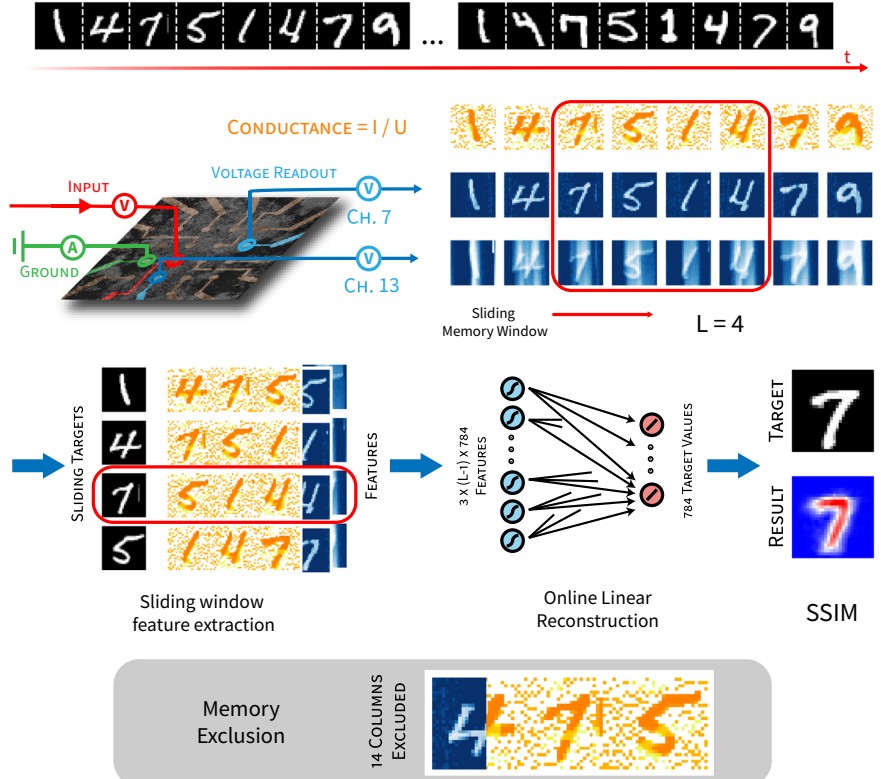

**Fig. 6 | Schematic outline of the sequence memory task.** Samples of a semi-repetitive 8-digit sequence (14751479) constructed from the MNIST dataset are temporally streamed into the NWN device through one input channel. A memory window of length $L$ ($L = 4$ shown as an example) slides through each digit of the readouts from 2 channels (7 and 13) as well as the network conductance. In each sliding window, the first (earliest) digit is selected for recall as its image is reconstructed from the voltage of one readout channel (channel 7) and memory features embedded in the conductance time series of later $L − 1$ digits in the memory window. Linear reconstruction weights are trained by the online learning method and reconstruction quality is quantified using the structural similarity index measure (SSIM). The shaded grey box shows an example of the memory exclusion process, in which columns of conductance memory features are replaced by voltage readouts from another channel (channel 13) to demonstrate the memory contribution to image reconstruction (recall) of the target digit.

jump from $L = 4$ to $L = 5$ (see also Fig. 8b). This reflects the contextual information used in the reconstruction: for $L = 4$, '9' is reconstructed from the sub-sequence '147', which is the same sub-sequence for '5', but for $L = 5$, '9' is uniquely reconstructed from sub-sequence '1475', with a corresponding increase in SSIM. This is not observed for digit '5'; upon closer inspection, it appears that the reconstruction of '5' suffers interference from '9' (see Supplementary Fig. S12) due to the common sub-sequence '147' and to the larger variance of '5' in the MNIST dataset (which also contributes to its misclassification). A similar jump in SSIM is evident for the repeat digit '7' from $L = 2$ to $L = 3$. For $L = 3$, the first instance of '7' (green curve) is reconstructed from '51', while the second instance (pink curve) is reconstructed from '91', so the jump in SSIM from $L = 2$ may be attributed to digit '7' leaving more memory traces in digit '1', which has a simpler structure than either '9' or '5'.

While the SSIM curves for each individual digit in the sequence increase only gradually with memory window length, their average (shown in Supplementary Fig. S11) shows an increase up to $L = 5$, followed by saturation. This reflects the repetition length of the sequence.

Figure 8c shows the maximum SSIM, averaged over all reconstructed digits using $L = 4$ when memory is increasingly excluded from the online reconstruction. SSIM decreases as more columns of conductance features are excluded (replaced with memoryless voltage features). This demonstrates that the memory embedded in the conductance features enhances online learning by the reconstructor. In particular, the maximum SSIM plateaus when ~28 and ~56 columns (corresponding to whole digits) are excluded and decreases significantly when the number of columns excluded is approximately 14,

42 or 70, indicating most of the memory traces are embedded in the central image pixels.

## Discussion

This study is the first to perform the MNIST handwritten digit classification benchmark task using an NWN device. In a previous study, Milano et al.[65] simulated an NWN device and mapped the readouts to a ReRAM cross-point array to perform in materia classification (with a 1-layer neural network) of the MNIST digits, achieving an accuracy of 90.4%. While our experimental implementation is different, readouts from their simulated NWN device also exhibited diverse dynamics and distinct states in response to different digit inputs, similar to that observed in this study. Other studies using memristor cross-bar arrays as physical reservoirs achieved lower MNIST classification accuracies[74,75]. In contrast, NWN simulation studies achieved higher classification accuracies of ≃98% by either pre-processing the MNIST digits with a convolutional kernel[62] or placing the networks into a deep learning architecture[76].

In this study, the relatively high classification accuracy achieved with online learning (93.4%) can be largely attributed to the iterative algorithm, which is based on recursive least squares (RLS). Previous RC studies by Jaeger et al.[77,78] suggested that RLS converges faster than least mean squares (similar to gradient-based batch methods), which tends to suffer more from numerical roundoff error accumulation, whereas RLS converges in a finite number of steps and uses the remaining training samples for fine-tuning[69]. This is evident in our results showing incremental learning of the weight matrix and is also corroborated by our mutual information analysis. While we performed online classification in an external digital layer, it may be possible to

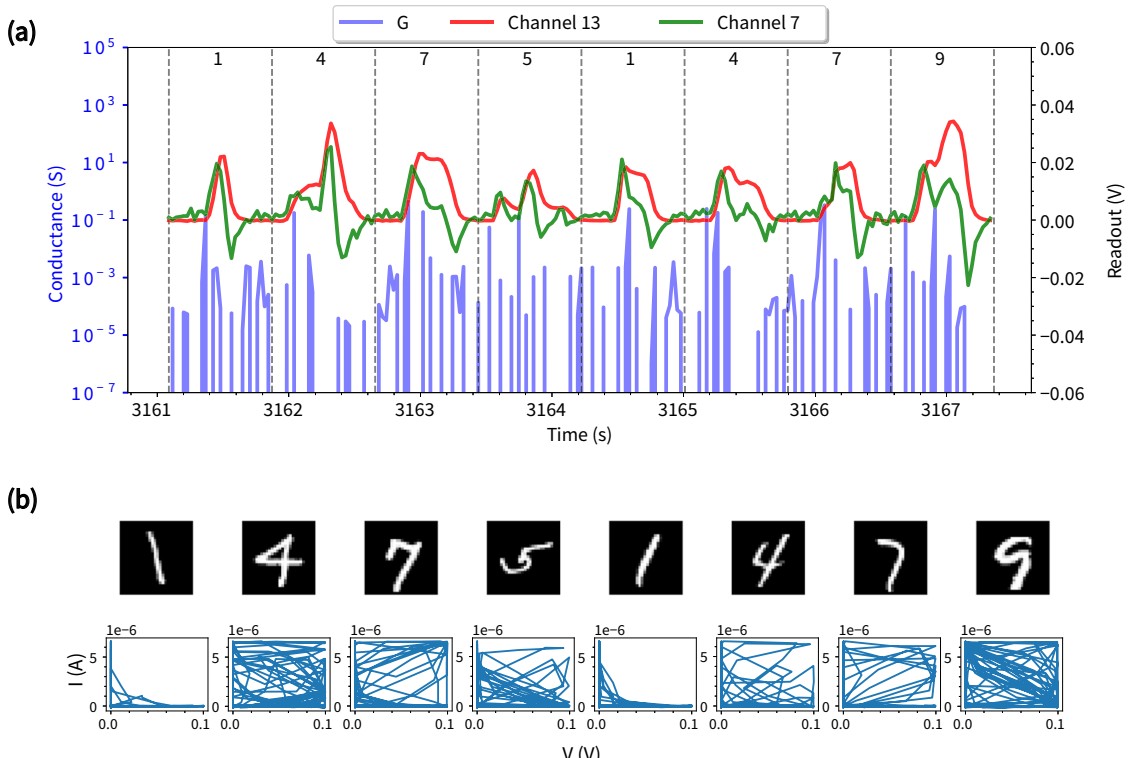

**Fig. 7 | NWN sequence memory. a** Conductance time series (G) and readout voltages for one full sequence cycle during the sequence memory task. For better visualisation, voltage readout curves are smoothed by averaging over a moving window of length 0.05 s and values for channel 7 are magnified by ×10. **b** Corresponding digit images and memory patterns in $I - V$ phase space.

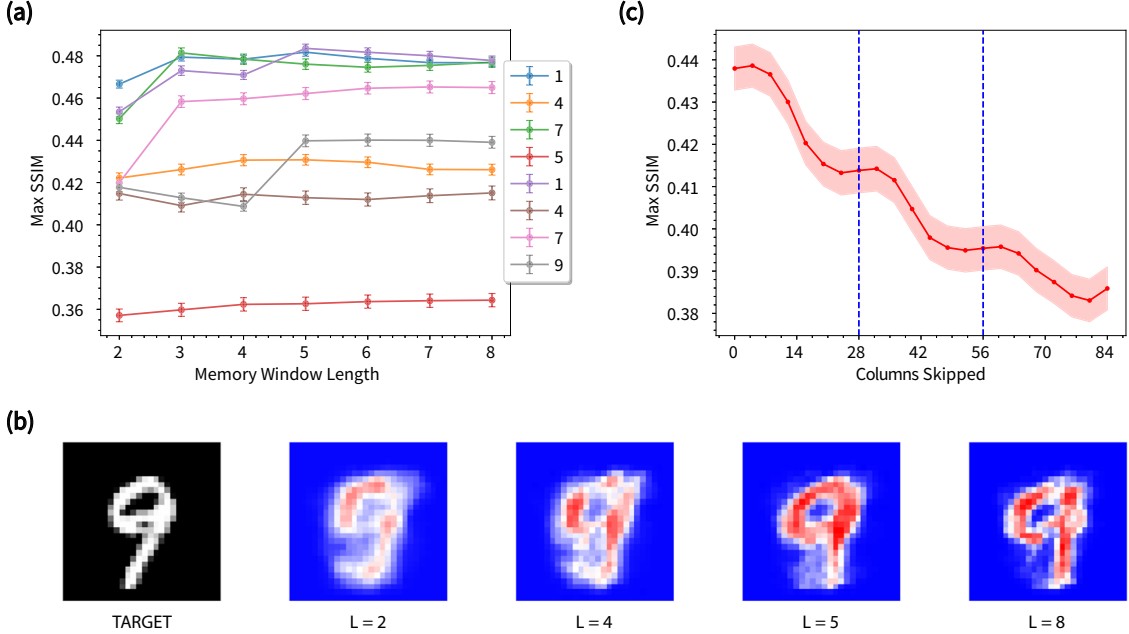

**Fig. 8 | Digit reconstruction in sequence memory task using the online learning algorithm. a** Maximum SSIM for each digit in the sequence as a function of the memory window length after the network learned from 7000 training sets. The testing set is comprised of 500 sequence samples. Error bars indicate the standard error of the mean across the samples within each digit class. **b** An example of a target digit from the MNIST dataset and the reconstructed digits using memory windows of different lengths. **c** Maximum SSIM with respect to the number of columns excluded in the memory feature space. The results are averaged across all testing digits using $L = 4$, and the standard error of the mean is indicated by the shading. Dashed blue lines indicate when whole digits are excluded.

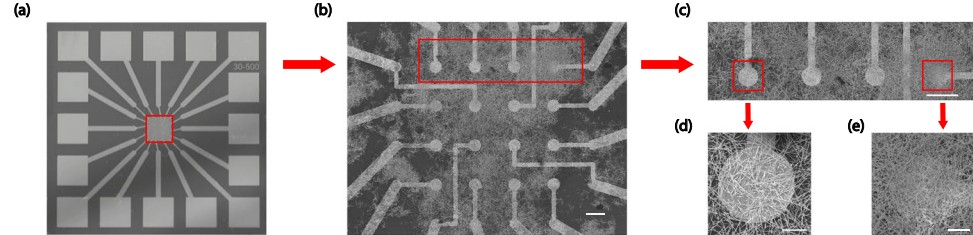

**Fig. 9 | Images of NWN device. a** Optical image of the multi-electrode array. Input/output are enabled by the outer electrodes. **b** Scanning electron microscopy (SEM) image of the $Ag_2Se$ Network. 16 inner electrodes are fabricated as a $4 \times 4$ grid and the nanowires are drop-casted on top of them. Scale bar: 100 μm. **c** Zoom-in of the SEM image for electrodes 0-3. Scale bar: 100 μm. **d** Zoom-in for electrode 0. Scale bar: 20 μm. **e** Zoom-in for electrode 3. Scale bar: 20 μm.

implement the online learning scheme in hardware using, for example, a cross-point array of regular resistors, which exhibit a linear (i.e., Ohmic) response. Such a system would then represent an end-to-end analogue hardware solution for efficient online dynamical learning in edge applications[29,79]. An all-analogue RC system was recently demonstrated by Zhong et al.[30] using dynamic resistors as a reservoir and an array of non-volatile memristors in the readout module.

Other studies have exploited the structure of memristor cross-bar arrays to execute matrix–vector multiplication used in conventional machine learning algorithms for MNIST classification, both in experiment[80,81] and simulation[82,83], although crosstalk in memristor cross-bars limits the accuracy of classification implemented in this type of hardware[80].

Beyond physical RC, unconventional physical systems like NWNs could potentially be trained with backpropagation to realise more energy-efficient machine learning than is currently possible with existing software and hardware accelerator approaches[84]. Furthermore, a related study by Loeffler et al.[85] (see also refs. [86,87]) demonstrates how the internal states of NWNs can be controlled by external feedback to harness NWN working memory capacity and enable cognitive tasks to be performed.

Information-theoretic measures like mutual information (MI) have been widely used to assess the intrinsic dynamics in random Boolean networks[88,89], Ising models[90], and the learning process of echo state networks[91] as well as artificial neural networks (ANNs)[92]. In a previous simulation study[62,64], we found that transfer entropy and active information storage in NWNs reveal that specific parts of the network exhibit richer information dynamics during learning tasks, and we proposed a scheme for optimising task performance accordingly. However, such element-wise calculations are not feasible for physical NWN hardware devices because the number of readouts from the system is limited by the size of the MEA. In this study, we applied a similar approach to that used in Shine et al.[92] to estimate the information content of ANNs at different stages during the MNIST classification task. They found unequal credit assignment, with some image pixels, as well as specific neurons and weights in the ANN, contributing more to learning than others. In our case, by investigating the information content embedded in the NWN readouts, we found that the learning process synchronises with the information provided by the dataset in the temporal domain, while each readout channel provides distinct information about different classes. Interestingly, we also observed some indication of channel preference for a specific digit class, which could potentially be further exploited for channel-wise tuning for other learning tasks.

The sequence memory task introduced in this study is novel and demonstrates both online learning and sequence memory recall from the memory patterns embedded in NWN dynamics. In the brain, memory patterns are linked with network attractor states[93]. The brain's neural network is able to remember sequence inputs by evolving the internal states to fixed points that define the memory pattern for the sequence[94]. In this study, we also found basins of attraction for the individual digits in the sequence, which allowed us to reconstruct the target digit image as a way of recalling the associated memory pattern. Delayed dynamics similar to that observed in the conductance time series of NWNs were also utilised by Voelker et al.[95] to build spiking recurrent neural networks[96] and implement memory-related tasks. In their studies, the delayed dynamics and memory are implemented in software-based learning algorithms, while NWNs are able to retain memory in hardware due to the memristive junctions and recurrent structure[33]. A similar study by Payvand et al.[97] demonstrated sequence learning using spiking recurrent neural networks implemented in ReRAM to exploit the memory property of this resistive memory hardware. Although their sequence was more repetitive than ours and task performance is measured differently, they demonstrated improved performance when network weights were allowed to self-organise and adapt to changing input, similar to physical NWNs. Future potential applications like natural language processing and image analysis may be envisaged with NWN devices that exploit their capability of learning and memorising dynamic sequences. Future computational applications of NWNs may be realised under new computing paradigms grounded in observations and measurements of physical systems beyond the Turing Machine concept[98].

In conclusion, we have demonstrated how neuromorphic nanowire network devices can be used to perform tasks in an online manner, learning from the rich spatiotemporal dynamics generated by the physical neural-like network. This is fundamentally different from data-driven statistical machine learning using artificial neural network algorithms. Additionally, our results demonstrate how online learning and recall of streamed sequence patterns are linked to the associated memory patterns embedded in the spatiotemporal dynamics.

## Methods
### Experimental setup
An NWN device, as shown in Fig. 9, was fabricated and characterised following the procedure developed in our previous studies[34–36,49,59]. Briefly, a multi-electrode array (MEA) device with 16 electrodes (4x4 grid) was fabricated as the substrate of the device using photo-lithographically patterned Cr/Ti (5 nm) and Pt (150 nm). Selenium nanowires were first formed by hydrothermal reduction of sodium selenite. $Ag_2Se$ nanowires were then synthesised by redispersing Se nanowires in a solution of silver nitrate ($AgNO_3$). The resulting nanowire solution was drop-casted over the inner electrodes of the MEA to synthesise the nanowire network (See Supplementary Fig. S1 for SEM images of the NWN without electrodes and Supplementary Fig. S2 for a simulated NWN as well as its corresponding graph representation). A data acquisition device (PXI-6368) was employed to deliver electric signals to the network and simultaneously read out the voltage time series from all electrodes. A source measurement unit (PXI-4141) was used to collect the current time series through the grounded electrode. A switch matrix (TB-2642) was used to modularly route all signals through the network as desired. All the equipment listed above was from National Instruments and controlled by a custom-made

LabView package designed for these applications[49,59,99]. The readout voltage data exhibited non-uniform phase shifts of $10 - 100\Delta t$ compared to the input stream, so a phase correction method was applied to prepare the readout data for further usage (see details in the following section).

## Online learning

Learning tasks were performed under a reservoir computing (RC) framework[58,59,62,99]. With $N$ digit samples used for training, the respective pixel intensities were normalised to $[0.1, 1]$ V as input voltage values and denoted by $\mathbf{U} \in \mathbf{R}^{N \times 784}$ for future reference. $\mathbf{U}$ was then converted to a 1-D temporal voltage pulse stream and delivered to an input channel while another channel was grounded. Each voltage pulse occupied $\Delta t = 0.001$ s in the stream. Voltage features were read simultaneously from $M$ other channels on the device (see Supplementary Fig. S3 for device setups). These temporal readout features were normalised and re-arranged to a 3-D array, $\mathbf{V} \in \mathbf{R}^{N \times M \times 784}$.

The phase of the readout voltage data ($V$) was adjusted per instance in the dataset based on the corresponding input using cross-correlation[100]. For the $n$-th digit sample, the respective segment in the input pulse stream was denoted as $\mathbf{u}_n \in \mathbf{R}^{784 \times 1}$, and the corresponding dynamical features from $M$ readout channels were represented by $[\mathbf{v}_{n,1}, \mathbf{v}_{n,2}, \ldots, \mathbf{v}_{n,m}]$, where $\mathbf{v}_{n,m} \in \mathbf{R}^{784 \times 1}$. The cross-correlation of $\mathbf{u}_n$ and $\mathbf{v}_{n,m}$ is calculated as:

$$C_{n,m}(\tau) = (\mathbf{u}_n * \mathbf{v}_{n,m})(\tau) = \sum_{t=1}^{784} \mathbf{u}_n(t)\mathbf{v}_{n,m}(t - \tau), \tag{1}$$

for $\tau = -783, -782, \ldots 0, \ldots, 783$. The phase difference $\phi$ is determined by:

$$\phi_{n,m} = \underset{\tau}{\mathrm{argmax}}(C_{n,m}(\tau)). \tag{2}$$

The 1-D phase adjustment was applied to the readout feature $\mathbf{v}_{n,m}$ of the instance based on the phase difference $\phi_{n,m}$.

The NWN device readouts embed dynamical features that are linearly separable, so classification can be performed in a linear output layer:

$$\mathbf{WA} = \mathbf{Y} \tag{3}$$

where $\mathbf{W}$ is the weight matrix (i.e., classifier), $\mathbf{A}$ is the readout feature space and $\mathbf{Y}$ contains the sample classes. An online method was implemented based on Greville's iterative algorithm for computing the pseudoinverse of linear systems[101]. This method is also a special case of the recursive least square (RLS) algorithm[69], using a uniform sample weighting factor of $\lambda = 1$.

The sample feature space was denoted by $\mathbf{A} = [\mathbf{a}_1, \mathbf{a}_2, \ldots, \mathbf{a}_n]$, $\mathbf{A} \in \mathbf{R}^{K \times N}$, in which each column ($\mathbf{a}_n$) represented one sample and every sample was composed of $K$ features ($K = 784M$). The corresponding classes for each sample were $\mathbf{Y} = [\mathbf{y}_1, \mathbf{y}_2, \ldots, \mathbf{y}_n]$, $\mathbf{Y} \in \mathbf{R}^{10 \times N}$. The order of the columns in $\mathbf{A}$ and $\mathbf{Y}$ are randomly shuffled to smooth the learning curve. During training, a new feature vector of the $n$-th digit sample $\mathbf{a}_n$ and its corresponding class vector $\mathbf{y}_n$ were appended to the right of respective matrices $\mathbf{A}_n$ and $\mathbf{Y}_n$ as columns, and the algorithm solved eqn. (3) for $\mathbf{W}(\mathbf{W} \in \mathbf{R}^{10 \times K})$ incrementally. The difference between the target $\mathbf{y}_n$ and the projected result using the previous weight matrix $\mathbf{W}_{n-1}$ was described by:

$$\mathbf{e}_n = \mathbf{y}_n - \mathrm{softmax}(\mathbf{W}_{n-1}\mathbf{a}_n). \tag{4}$$

When $\|\mathbf{e}_n\|$ was below a preset threshold $e' = 0.1$, $\mathbf{W}$ was updated by:

$$\mathbf{W}_n = \mathbf{W}_{n-1} + \mathbf{e}_n\mathbf{b}_n^T, \tag{5}$$

where

$$\mathbf{b}_n = \frac{\boldsymbol{\theta}_{n-1}\mathbf{a}_n}{1 + \mathbf{a}_n^T \boldsymbol{\theta}_{n-1}\mathbf{a}_n}, \tag{6}$$

with

$$\boldsymbol{\theta}_n = \boldsymbol{\theta}_{n-1} - \boldsymbol{\theta}_{n-1}\mathbf{a}_n\mathbf{b}_n^T. \tag{7}$$

For the cases when $\|\mathbf{e}_n\|$ was above the threshold, an error-correction scheme was applied to optimise the result[68]. In addition, $\mathbf{A}$, $\mathbf{Y}$ and $\boldsymbol{\theta}$ were initialised at $n = 0$ by:

$$\mathbf{A}_0 = \epsilon\mathbf{I}, \qquad \mathbf{Y}_0 = \mathbf{0}, \qquad \text{and} \qquad \boldsymbol{\theta}_0 = \frac{1}{\epsilon^2}\mathbf{I}, \tag{8}$$

with $\epsilon = \overline{|\mathbf{A}|}$.

## Mutual information

To gain deeper insight into the network's behaviour and attribute real-time learning to its dynamics, mutual information (MI) between the dynamical features and the corresponding classes was calculated to estimate the information content in a way similar to a previous study on ANNs[92]. All MI results were calculated using the Java Information Dynamics Toolkit (JIDT)[102]. MI was estimated spatially based on the pixel positions from different readout channels and temporally as the feature space expanded when more samples were learned. Among the $N$ digit samples delivered to the network, an ensemble was created using the readout data from channel $m$ at the $i$-th pixel position: $\mathbf{V}_{m,i} = [v_{1,m,i}, v_{2,m,i}, \ldots, v_{N,m,i}]$, $\mathbf{V}_{m,i} \in \mathbf{R}^{1 \times N}$. Another class vector $\mathbf{P} \in \mathbf{R}^{1 \times N}$ was created and mutual information was estimated accordingly by:

$$\mathcal{M} = \Omega_{\mathrm{MI}}[\mathbf{V}_{m,i}, \mathbf{P}] \tag{9}$$

where $\Omega_{\mathrm{MI}}$ stands for the mutual information operator, where the Kraskov estimator was employed[103].

A 3-D matrix $\mathcal{M} \in \mathbf{R}^{N \times M \times 784}$ was generated after calculating spatial-temporally throughout $\mathbf{V}$. $\mathcal{M}$ was averaged across the pixel axis (third) to obtain the temporal mutual information per channel. The spatial analysis of mutual information was based on the calculation result for the whole dataset. The class-wise interpretation of $\mathcal{M}$ was generated by averaging across samples corresponding to each digit class.

## Sequence memory task

A sequence-based memory task was developed to investigate sequence memory and recall. Samples of an 8-digit sequence with a semi-repetitive pattern (14751479) were constructed by randomly sampling the respective digits from the MNIST dataset. Input pixel intensities were normalised to the range $[0, 0.1]$ V, and the samples were streamed into and read out from the NWN in the same way as the classification task, using channels 9, 8 and 7 for input, ground and readout, respectively. In addition to dynamical features from the voltage readouts, memory features were used from the network conductance, calculated pixel-wise by

$$G_i = \frac{\mathcal{I}_i}{U_i}, \tag{10}$$

where $\mathcal{I}$ is the current leaving the ground channel and $\mathbf{U}$ is the input voltage.

To test recall, a digit from the sequence was selected and its image was reconstructed from voltage readouts and memory features in the conductance time series corresponding to digits later in the sequence. A variable memory window of length $L \in [2, 8]$ determines the

sequence portion used to reconstruct a previous digit image, i.e., from $L-1$ subsequent digits. For example, a moving window of length $L=4$ reconstructs the first (target) digit from the conductance memory features in the subsequent 3 digits (cf. Fig. 6). By placing the target digits and memory features into ensembles, a dataset of 7000 training samples and 500 testing samples was composed using the sliding windows.

To reconstruct each target digit image, the same linear online learning algorithm used for MNIST classification was applied. In this case, $\mathbf{Y}$ in eqn. (3) was composed as $\mathbf{Y}=[\mathbf{y}_1, \mathbf{y}_2, \ldots, \mathbf{y}_n]$, with $\mathbf{Y} \in \mathbf{R}^{784 \times N}$, and softmax in eqn. (4) was no longer used. Structural similarity index measure (SSIM)[104] was employed to quantify the reconstruction quality.

To further test that image reconstruction exploits memory features and not just dynamical features associated with the spatial pattern of the sequence (i.e., sequence classification), a memory exclusion test was developed as follows. The conductance features corresponding to a specified number of columns of inputs were replaced by voltage features from channel 13 (voltages are adjusted to the same scale as conductance) so that the memory in conductance is excluded without losing the non-linear features in the readout data (cf. Fig. 6). The target digit was then reconstructed for a varying number of columns with memory exclusion.

## Data availability
The raw NWN measurement data used in this study are available in the Zenodo database https://zenodo.org/record/7662887.

## Code availability
The code used in this study is available in the Zenodo database https://zenodo.org/record/7662887.

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

## Acknowledgements
The authors wish to thank members of the UCLA Nanofabrication Laboratory and the California NanoSystems Institute (CNSI) Nano and Pico Characterization Lab (NPC) for their support of this project. The authors also acknowledge the use of the Artemis High-Performance Computing resource at the Sydney Informatics Hub, a Core Research Facility of the University of Sydney. R.Z. is supported by a Postgraduate Research Excellence Award scholarship from the University of Sydney. A.L. is supported by a Research Training Program scholarship from the University of Sydney. Z.K. acknowledges support from the Australian-American Fulbright Commission.

## Author contributions
R.Z. and Z.K. conceived and designed the study. S.L., A.Z.S. and J.G. fabricated the device. S.L. performed the experiments with guidance from R.Z., A.Z.S., J.G. and Z.K. R.Z., A.L., J.L. and Z.K. analysed the data. R.Z. wrote the manuscript with consultation from the other authors. Z.K. supervised the project.

## Competing interests
Z.K., A.Z.S. and J.G. are with Emergentia, Inc. The authors declare no other competing interests.
