## [Peer Review File · Nature Communications]

REVIEWER COMMENTS

Reviewer #1 (Remarks to the Author):

The paper reports and analyzes the hardware demonstration for classification/learning of MNIST digits with nano wire networks (NWN) using 16 devices.

My main comment is the following:

1- The way the classification and learning is setup in the experiment is not considered online learning/inference task. An “online” for the temporal task is one where the inputs are “streaming” and the classification and learning is done in real time. If this was the case, the network should have had 784 ms of memory, based on the way the authors have encoded the MNIST digits. Each pixel is presented for 1 ms with a certain voltage pulse which is a function of the intensity of the pixel (the intensity is binarized). At the end of the presentation of each digit (which is at least 784 ms), the network needs to classify/learn the digit. Does this network have 784 ms of short-term memory? This seems unlikely because of the size of the network, and lack of recurrency between the nodes. If the entire sequence is saved in a matrix of $784 \times m$ (number readouts), that indicates that an external memory is being used for this and hence this is not a real-time task.

2- The set up in figure 1 does not show any recurrency between the nodes. The lack of recurrency between the nodes means that the network does not constitute a reservoir computing network, but more of Extreme Learning Machines (ELMs), where the devices are only used as a feedforward random projection. Even in the case of ELM, the dimensionality of the hidden layer is very small (16 neurons) and the theory of projecting to higher dimensional space does not really apply here. Can the authors explain why with such low dimensionality they claim that the digits become linearly separable? I do not think this sentence in page 5 of the main document is convicting enough: “Most importantly, it is evident that the readouts for each group of samples (digits ‘0’ and ‘5’) are distinct from each other. This demonstrates that the nanowire network nonlinearly maps the input signals into a higher- dimensional space containing linearly separable features.” Just because the readout is distinct between the samples of one class does not indicate the non-linear projections and it definitely does not indicate linear separability.

3- The comparison table 1 shows the difference between learning using a hidden layer and without hidden layer, and is not necessarily the difference between online and offline case. The classifier

labeled as “online” in the table, is the result of the classifier which is read from the nano-wire network, and the one labeled as batch is the result of the “fully connected, 1-layer artificial neural network”. Therefore, the difference is not because of offline vs online, but because in the former, the network has one extra layer. This should be clarified in the text. Also, what is the meaning of 5 readout in the “batch” case? Since there is no hidden layer, where does “5” come from?

4- The term “incremental learning” is a reserved word for the continual learning literature that proposes a solution for “catastrophic forgetting” problem in Machine learning, and it does not apply here. So please do not use this term in page 6.

5- It is not clear if the accuracy numbers reported in table 1 and figure 3 are from test set or training set?

6- In Fig. 4b what is the y axis? Why are there 13 channels? Wasn't there 5 channels? The caption of Fig. 4 should be more informative. In the text it says that fig. 4b presents the MI between the 10 MNIST digits and each NWN device readouts. But in the caption there is a reference to input channel 0. I find this confusing. Please elaborate and explain better.

7- Comment 1 and 2 also applies to the sequence memory task.

8- Minor comment: the references are not sequentially ordered.

Reviewer #2 (Remarks to the Author):

The authors considered the random network as a neuromorphic system. They performed online learning from spatiotemporal dynamical features using MNIST character image classification and sequence memory recall tasks with reservoir operations. They argue that there is a correlation between the accuracy of individual digit classification and the amount of mutual information.

The experimental procedures and considerations seem correct from the results obtained with high originality, but the data from a single sample is not encouraging. Proceed to increase the number of measured devices or add additional experiments and discussions that cover them.

I would like to make the article published after major revision.

p6 Table 1 and Fig. 3: The authors discuss a very small difference in classification accuracy between the channel one and channel 5 cases. It seems meaningless because it is feared that such a difference could easily be reversed if the parameters and calculation methods were slightly changed. I agree that there is a 5% difference between batch and online calculations and that significance exists. This is certainly a correct assessment of one device. However, different devices cannot have the same arrangement of nanowires. If you bring different devices, the results are likely to be different. The authors need to clarify why such an argument can be made for a small difference in the system due to the difference in the number of channels.

Miscellaneous

Abstract

p1 Lastline: The word “nano–structured” should be “nanostructured”.

p2 L6: The word “spatio-temporal” should be “spatiotemporal”. (same below as well)

p2 L8: The words “a NWN” should be “an NWN”.

NCOMMS-23-10545 Author Responses to Reviewer Comments

We appreciate the reviewers' constructive comments and feedback, all of which are addressed below in a point-by-point manner. We have revised our manuscript accordingly.

Reviewer #1 (Remarks to the Author):

The paper reports and analyzes the hardware demonstration for classification/learning of MNIST digits with nano wire networks (NWN) using 16 devices.

My main comment is the following:

1- The way the classification and learning is setup in the experiment is not considered online learning/inference task. An "online" for the temporal task is one where the inputs are "streaming" and the classification and learning is done in real time. If this was the case, the network should have had 784 ms of memory, based on the way the authors have encoded the MNIST digits. Each pixel is presented for 1 ms with a certain voltage pulse which is a function of the intensity of the pixel (the intensity is binarized). At the end of the presentation of each digit (which is at least 784 ms), the network needs to classify/learn the digit. Does this network have 784 ms of short-term memory? This seems unlikely because of the size of the network, and lack of recurrency between the nodes. If the entire sequence is saved in a matrix of 784 x m (number readouts), that indicates that an external memory is being used for this and hence this is not a real-time task.

RESPONSE:

The nanowire network used in this work is composed of thousands of nanowires (neurons) and several times more memristive junctions (synapses), which necessarily makes it a recurrent network with short-term working memory capacity. However, the network does not classify/learn the digits. Rather, it encodes their dynamical features. Classification of the extracted dynamical features is performed by an external layer (which is digital, so uses 784 x m external digital memory) using an iterative algorithm based on recursive least squares (RLS), which is a well-known online machine learning method for solving the least squares problem in an online manner¹⁻³. Because the digits are continuously streamed into the device, and their dynamical features continuously extracted from the device, and because the features are classified sample-by-sample using an online machine learning method, we have described this as "online dynamical learning".

In the revised manuscript, we have improved the contrast in Figs. 1 and 6 to make the nanowires more visible against the electrodes in the SEM images:

Fig. 1

Fig. 6:

In the revised manuscript, we have added Supplementary Figure S1, which shows SEM images of the nanowire network without the electrodes.

Supplementary Figure S1. SEM images of the nanowire network without electrodes. Scale bars are: (a) 10 μm ; and (b) 5 μm .

Additionally, we have added Supplementary Figure S2, which shows an example of a simulated nanowire network and its graphical representation, based on a model published in our previous studies⁴⁻⁷, to better visualize the recurrent structure of the network. Here, nodes represent nanowires and edges represent memristive junctions. We have used this model previously to simulate reservoir computing tasks⁶⁻¹².

Supplementary Figure S2. Simulated nanowire network. (a) Network of 1024 nanowires with centres and lengths drawn from a uniform and Gamma distribution, respectively. (b) Corresponding graph representation, with nanowires represented by nodes and memristive cross-points between nanowires represented by edges (6877 total edges). Simulations are based on a physically-motivated model from previous studies [1-4].

2- The set up in figure 1 does not show any recurrency between the nodes. The lack of recurrency between the nodes means that the network does not constitute a reservoir computing network, but more of Extreme Learning Machines (ELMs), where the devices are only used as a feedforward random projection. Even in the case of ELM, the dimensionality of the hidden layer is very small (16 neurons) and the theory of projecting to higher dimensional space does not really apply here. Can the authors explain why with such low dimensionality they claim that the digits become linearly separable? I do not think this sentence in page 5 of the main document is convicting enough: "Most importantly, it is evident that the readouts for each group of samples (digits '0' and '5') are distinct from each other. This demonstrates that the nanowire network nonlinearly maps the input signals into a higher- dimensional space containing linearly separable features." Just because the readout is distinct between the samples of one class does not indicate the non-linear projections and it definitely does not indicate linear separability.

RESPONSE:

Continuing from our previous response, the nanowire neurons constitute a recurrent network that serves as the hidden layer and projects the input signals into a higher-dimensional dynamical feature space (cf. Takens's embedding theorem¹³). The 16 electrode neurons serve as the electronic interface for signal input and read out of the features from different parts of the heterogeneous network. To further highlight the recurrent structure of the network, we have also added Supplementary Figures S1 and S2 (cf. our response to the previous comment). Our previous studies of similar nanowire networks have demonstrated the complex recurrent structure of the network¹⁴⁻¹⁶ and how this property can be harnessed to perform various reservoir computing tasks^{8,16,17}. Similarly, studies by other groups¹⁸ have also demonstrated reservoir computing with a multi-electrode nanowire network device.

Note that physical reservoir computing implemented in these devices differs from traditional algorithmic reservoir computing in several ways: 1. the memristive edges are not programmable, meaning that their weights cannot be assigned arbitrary fixed values like those in algorithmic reservoirs; 2. the input signal is delivered to only one electrode (which is in contact with a small, localised subset of nanowire nodes), whereas in algorithmic reservoirs, the input signal is mapped to all nodes using an input weight matrix; and 3. nanowire networks display a heterogeneous topology, with small-worldness, hubs and modularity¹⁹, which differs from random networks used as algorithmic reservoirs. These differences notwithstanding, the fact that we achieve a relatively high accuracy using linear classification demonstrates that the features produced by the network are linearly separable to a relatively high degree. If the feature embedding was nonlinear, task performance would be considerably worse using linear classification and instead a clustering algorithm would have to be used.

In the revised manuscript, we have improved the visibility of inter-class features in Figure 2, which now shows channel readouts for the 2 selected digits, averaged over 100 samples of each.

Fig. 2. Input–output mapping of MNIST digits in a nanowire network device. Images of MNIST digits ‘0’ and ‘5’ averaged across 100 samples randomly selected from the training set (column 1), their corresponding input voltage streams (column 2, red) and readout voltages from multiple channels (columns 3-7, blue).

Additionally, we have included Supplementary Figures S4 and S5, which show this for all 10 classes and all channel readouts.

Supplementary Figure S4. Readout voltages for digits ‘0’ - ‘4’ from all 16 channels. Channel 0 and 3 are used for input and drain respectively. For each digit class, readouts correspond to 100 samples randomly selected from the training set and averaged for each channel.

Supplementary Figure S5. Readout voltages for digits ‘5’ - ‘9’ from all 16 channels. Channel 0 and 3 are used for input and drain respectively. For each digit class, readouts correspond to 100 samples randomly selected from the training set and averaged for each channel.

3- The comparison table 1 shows the difference between learning using a hidden layer and without hidden layer, and is not necessarily the difference between online and offline case. The classifier labeled as “online” in the table, is the result of the classifier which is read from the nano-wire network, and the one labeled as batch is the result of the “fully connected, 1-layer artificial neural network”. Therefore, the difference is not because of offline vs online, but because in the former, the network has one extra layer. This should be clarified in the text. Also, what is the meaning of 5 readout in the “batch” case? Since there is no hidden layer, where does “5” come from?

RESPONSE:

Both methods in Table 1 use the nanowire network as the hidden layer. The readout features from different channels are concatenated and delivered to two different classifiers, i.e. batch and online. The difference between these two classifiers is that the batch method uses gradient-based regression on the whole training dataset, while the online method uses recursive regression applied on a sample-by-sample basis. A detailed comparison of these two algorithms can be found in Jaeger’s study²⁰ as well as in chapter 12 and 13 in Farhang-Boroujeny et al.’s book².

In the revised manuscript, we have clarified this in Results, p.4, lines 2-4:

“The weights are learned from the dynamical features and updated after each digit sample using an online *iterative algorithm based on recursive least squares (RLS)*.”

And in Results, Online learning, p. 5, lines 7-13, as follows:

“Table 1 presents using the online method (*external weights trained by an RLS method*). batch method (*external weights trained by backpropagation with gradient descent*). *Both classifiers learn from the dynamical features extracted from the NWN, with readouts delivered to the two classifiers separately.*”

4- The term “incremental learning” is a reserved word for the continual learning literature that proposes a solution for “catastrophic forgetting” problem in Machine learning, and it does not apply here. So please do not use this term in page 6.

RESPONSE:

We have changed the expression in Results, Online Learning, p.5, lines 20-22:

“A key advantage of the online method is that *continuously learning from the streaming input data* enables relatively rapid convergence, as shown next.”

5- It is not clear if the accuracy numbers reported in table 1 and figure 3 are from test set or training set?

RESPONSE:

The accuracy numbers are calculated using samples from a separate testing set, which is composed of 1000 samples and has not been presented to the classifier before. This has now been clarified in the caption of Table 1.

Readout Data	Classifier	Accuracy
784 x 1	online	94.9%
784 x 5	online	96.2%
784 x 1	batch	92.2%
784 x 5	batch	93.8%

Table 1. MNIST digit classification using an NWN device with one and five readout channels for online and batch (offline) classifiers. Each classification accuracy is calculated for 1,000 testing samples and averaged across 5 iterations. The online classifier is trained for a single epoch of 50,000 training samples, while the batch-based classifier is trained and optimized using gradient descent for 100 epochs with 500 mini-batches of size 100.

6- In Fig. 4b what is the y axis? Why are there 13 channels? Wasn't there 5 channels? The caption of Fig. 4 should be more informative. In the text it says that fig. 4b presents the MI between the 10 MNIST digits and each NWN device readouts. But in the caption there is a reference to input channel 0. I find this confusing. Please elaborate and explain better.

RESPONSE:

There are 16 electrode channels (0-15) in total available in the system as shown in Fig. 1. We used 2 channels (0,3) for input and ground, respectively, with the others available for readout. For the MNIST classification task, we used up to 5 of them (1,2,13,15,12). The caption and axis labels of Fig. 4(b) have been updated in the revised manuscript as follows:

Fig. 4. Learning rate and mutual information. (a) Mean of the magnitude of changes in the linear weight matrix, $|\Delta W|$, as a function of number of samples learned by the network. (b) Corresponding Mutual Information (MI) for each of the 5 channels used for online classification (cf. Fig. 3) and for input channel 0.

The y-axis of Fig. 4b is channel index, sorted according to the order with which they are presented to the classifier (same as for Fig. 3(b)). The MI values visualized in Fig. 4(b) are averaged over 10 digit classes for each readout channel used in the task. This has also been clarified in Results, Online Learning, pg. 7, lines 14-17:

This is indicated by Fig. 4(b), which presents mutual information (MI) *between the 10 MNIST digit classes and each of the NWN device readouts* used for online classification (cf. Fig. 3).

7- Comment 1 and 2 also applies to the sequence memory task.

RESPONSE:

The sequence memory task is implemented in the same way as the MNIST classification task, except the electrodes used are different (see Supplementary Figure S3 for the electrode setup). In this task, we show that memory from previous inputs in the digit stream may be exploited for memory-based tasks. In general, nanowire networks demonstrate two types of memory: short-term (fading) memory facilitated by the recurrent structure, and long-term memory facilitated by the memristive junctions. The presence of memory traces can be clearly observed in the conductance shown in Fig. 7a, which highlights the potential of harnessing these memories for learning. Our responses to comments 1 and 2 are thus also relevant for the sequence memory task.

8- Minor comment: the references are not sequentially ordered.

RESPONSE:

We have checked this but cannot find where it occurs. Perhaps this may appear to be the case at first glance, since some references cited early on are cited more than once throughout the manuscript, so the reference numbers may appear to be out of sequence, whereas in fact they are not.

Reviewer #2 (Remarks to the Author):

The authors considered the random network as a neuromorphic system. They performed online learning from spatiotemporal dynamical features using MNIST character image classification and sequence memory recall tasks with reservoir operations. They argue that there is a correlation between the accuracy of individual digit classification and the amount of mutual information.

The experimental procedures and considerations seem correct from the results obtained with high originality, but the data from a single sample is not encouraging. Proceed to increase the number of measured devices or add additional experiments and discussions that cover them. I would like to make the article published after major revision.

p6 Table 1 and Fig. 3: The authors discuss a very small difference in classification accuracy between the channel one and channel 5 cases. It seems meaningless because it is feared that such a difference could easily be reversed if the parameters and calculation methods were slightly changed. I agree that there is a 5% difference between batch and online calculations and that significance exists. This is certainly a correct assessment of one device. However, different devices cannot have the same arrangement of nanowires. If you bring different devices, the results are likely to be different. The authors need to clarify why such an argument can be made for a small difference in the system due to the difference in the number of channels.

RESPONSE:

It is correct that the measurements presented are from a single device, but we performed a range of additional experiments involving different combinations of electrodes used for input, drain and readouts, as well as different encodings of the MNIST digits. In our experience, from previous studies using both physical^{8,17} and simulated⁶⁻¹² nanowire networks, we have found that unless the networks are extremely sparse or extremely dense, the statistical variance due to different nanowire networks has less of an impact on task performance than the systematic variations due to the device itself. This is because the electrodes are in contact with different parts of the network, which is highly heterogeneous, so using different combinations of electrodes for input/drain/readouts lead to variations in the nonlinear mappings for the same input. Additionally, there are systematic variations associated with whether the electrical contact made with the nanowires is good or poor.

In the revised manuscript, we have added Supplementary Figure S7 which compares MNIST classification accuracies using up to 7 channels with different electrode combinations and voltage encodings for input. The difference between (a) and (b) lies in the order with which the channel readouts are delivered to the online classifier. This demonstrates how different device configurations and operation can lead to varying performance. Note that an important implication of our results is that the online learning method is relatively robust against device variations (statistical and systematic), and this is why the differences in accuracies remain relatively small.

Supplementary Figure S7. MNIST online classification accuracy as a function of number of channels using different electrodes for (input,drain) and different voltage amplitude and bias. Maximum accuracy for 1,000 testing samples (after training on 50,000 samples) with respect to number of readout channels used by the online linear classifier. Error bars indicate the standard error of the mean of 5 measurements with different sequences of the training samples. (a) Source/drain electrodes (0,3), with channels 1,2,13,15,12,9,14 used for readout to the classifier; and source/drain electrodes (9,8), with channels 1,0,13,3,12,6,15 used for readout. (b) Same as (a), but with channels 12,7,14,1,0,13,3 used for (9,8). For all cases, readouts are streamed to the classifier in the specified order.

Miscellaneous

Abstract

p1 Lastline: The word “nano–structured” should be “nanostructured”.

p2 L6: The word “spatio-temporal” should be “spatiotemporal”. (same below as well)

p2 L8: The words “a NWN” should be “an NWN”.

RESPONSE:

These have been changed.

References

1. Engel, Y., Mannor, S. & Meir, R. The Kernel Recursive Least-Squares Algorithm. *IEEE Trans. Signal Process.* **52**, 2275–2285 (2004).
2. Farhang-Boroujeny, B. *Adaptive Filters: Theory and Applications*. (John Wiley & Sons, 2013).
3. Islam, S. A. U. & Bernstein, D. S. Recursive Least Squares for Real-Time Implementation [Lecture Notes]. *IEEE Control Syst. Mag.* **39**, 82–85 (2019).
4. Kuncic, Z. *et al.* Emergent brain-like complexity from nanowire atomic switch networks: Towards neuromorphic synthetic intelligence. in *2018 IEEE 18th International Conference on Nanotechnology (IEEE-NANO)* 1–3 (2018). doi:10.1109/NANO.2018.8626236.
5. Kuncic, Z. *et al.* Neuromorphic Information Processing with Nanowire Networks. in *2020 IEEE International Symposium on Circuits and Systems (ISCAS)* 1–5 (2020). doi:10.1109/ISCAS45731.2020.9181034.
6. Hochstetter, J. *et al.* Avalanches and edge-of-chaos learning in neuromorphic nanowire networks. *Nat. Commun.* **12**, 4008 (2021).
7. Zhu, R. *et al.* Information dynamics in neuromorphic nanowire networks. *Sci. Rep.* **11**, 13047 (2021).
8. Sillin, H. O. *et al.* A theoretical and experimental study of neuromorphic atomic switch networks for reservoir computing. *Nanotechnology* **24**, 384004 (2013).
9. Fu, K. *et al.* Reservoir Computing with Neuromemristive Nanowire Networks. in *2020 International Joint Conference on Neural Networks (IJCNN)* 1–8 (2020). doi:10.1109/IJCNN48605.2020.9207727.
10. Zhu, R. *et al.* Harnessing adaptive dynamics in neuro-memristive nanowire networks for transfer learning. in *2020 International Conference on Rebooting Computing (ICRC)* 102–106 (2020). doi:10.1109/ICRC2020.2020.00007.

11. Loeffler, A. *et al.* Modularity and multitasking in neuro-memristive reservoir networks. *Neuromorphic Comput. Eng.* **1**, 014003 (2021).
12. Zhu, R. *et al.* MNIST classification using Neuromorphic Nanowire Networks. in *International Conference on Neuromorphic Systems 2021* 1–4 (Association for Computing Machinery, 2021). doi:10.1145/3477145.3477162.
13. Takens, F. Detecting strange attractors in turbulence. in *Dynamical Systems and Turbulence, Warwick 1980* (eds. Rand, D. & Young, L.-S.) vol. 898 366–381 (Springer Berlin Heidelberg, 1981).
14. Avizienis, A. V. *et al.* Neuromorphic Atomic Switch Networks. *PLoS ONE* **7**, e42772 (2012).
15. Stieg, A. Z. *et al.* Emergent Criticality in Complex Turing B-Type Atomic Switch Networks. *Adv. Mater.* **24**, 286–293 (2012).
16. Demis, E. C. *et al.* Atomic switch networks—nanoarchitectonic design of a complex system for natural computing. *Nanotechnology* **26**, 204003 (2015).
17. Lilak, S. *et al.* Spoken Digit Classification by In-Materio Reservoir Computing With Neuromorphic Atomic Switch Networks. *Front. Nanotechnol.* **3**, (2021).
18. Milano, G. *et al.* In materia reservoir computing with a fully memristive architecture based on self-organizing nanowire networks. *Nat. Mater.* (2021) doi:10.1038/s41563-021-01099-9.
19. Loeffler, A. *et al.* Topological Properties of Neuromorphic Nanowire Networks. *Front. Neurosci.* **14**, 184 (2020).
20. Jaeger, H. A tutorial on training recurrent neural networks, covering BPPT, RTRL, EKF and the ‘echo state network’ approach. (2002).

REVIEWER COMMENTS

Reviewer #1 (Remarks to the Author):

I thank the authors for addressing my questions.

Importantly, the change of contrast in Fig. 1 and the added supplementary figures definitely help in understanding there is a mesh of NWN below the electrodes and hence the recurrent network!
Thanks!

However, for someone who is not familiar with NWNs, the manuscript is not necessarily self-contained. It assumes that the reader should know how the NWN can act as a recurrent network. I suggest writing a few introductory sentences. For me, before reading the other papers, it was not clear at all that the network has 1000s of neurons and synapses, and the electrodes are only sending signals in and out.

Now, having read reference 62 (<https://www.nature.com/articles/s41563-021-01099-9>

), the novelty of the work is not as significant, though it is a good engineering effort.

Reference 62 by Milano et al. which was published in nature material in 2021, performed a similar smaller task on the hardware, while performing the temporal MNIST on software. This paper basically has the same story and idea, and instead of simulating MNIST in software, it is now implementing it on hardware, with more data analysis. This is the main comment.

More detailed comments:

- It would be beneficial to add a few sentences about how the images are converted to 1D temporal signals. For example, each pixel value is converted to a pulse frequency in 1 ms.

- The analysis on the mutual information is clearly showing the correlation between the state of the substrate and the images. I am also curious to know the jump in the accuracy that is made as a result of the NWNs. For example, if you directly feed the Nx 784 inputs to the linear read-out of an SNN, what would be the accuracy?

- I don't understand the significance of table 1. The number of samples for convergence is the same. It is not clear why there is the claim that online learning can converge faster.

- In the response letter, the authors mention that this is different from reservoir computing as the computational graph resembles Small World connectivity. What's the significance of small-world connectivity in solving temporal tasks?

- How do the authors foresee the embedded system using NWNs? Would it be possible to have this whole system with the read-out, which now includes a microcontroller, in one system?

- How do the systems with NWNs generally predicted to scale? As the other reviewer also mentioned, how reliable is the computation from one device to another?

Reviewer #2 (Remarks to the Author):

By reading the revised article carefully, the authors revised well by following the reviewers' pointing out by adding new discussion with new data and supplements. I think the article now achieved the published level of nature communications.

NCOMMS-23-10545A Author Responses to Reviewer Comments V2

Reviewer #1 (Remarks to the Author):

I thank the authors for addressing my questions.

Importantly, the change of contrast in Fig. 1 and the added supplementary figures definitely help in understanding there is a mesh of NWN below the electrodes and hence the recurrent network! Thanks!

However, for someone who is not familiar with NWNs, the manuscript is not necessarily self-contained. It assumes that the reader should know how the NWN can act as a recurrent network. I suggest writing a few introductory sentences. For me, before reading the other papers, it was not clear at all that the network has 1000s of neurons and synapses, and the electrodes are only sending signals in and out.

RESPONSE:

An introductory sentence has been added on p.2 line 32-33, as follows:

“Typically, each NWN contains thousands of nanowires and even greater number of junctions.”

Now, having read reference 62 (<https://www.nature.com/articles/s41563-021-01099-9>), the novelty of the work is not as significant, though it is a good engineering effort.

Reference 62 by Milano et al. which was published in nature material in 2021, performed a similar smaller task on the hardware, while performing the temporal MNIST on software. This paper basically has the same story and idea, and instead of simulating MNIST in software, it is now implementing it on hardware, with more data analysis. This is the main comment.

RESPONSE:

The novelty of our work, relative to the previous work mentioned above, is twofold:

1. We implemented an online learning algorithm for the reservoir output layer which enables higher training efficiency and higher classification accuracy;
2. We developed a complex sequence memory task using the MNIST digits and showed how memory enhances learning. This has never been done before.

More detailed comments:

- It would be beneficial to add a few sentences about how the images are converted to 1D temporal signals. For example, each pixel value is converted to a pulse frequency in 1 ms.

RESPONSE:

This is described in Methods, online learning, page 15, lines 18-22, however, we have now also added a clarifying sentence to the caption of Figure. 1:

“MNIST handwritten digit samples (N samples \times 784 pixel features) are normalised and converted to 1-D temporal voltage pulse streams (each pixel occupies $\Delta t = 0.001$ s) and delivered consecutively to the nanowire multi-electrode device.”

- The analysis on the mutual information is clearly showing the correlation between the state of the substrate and the images. I am also curious to know the jump in the accuracy that is made as a result of the NWNs. For example, if you directly feed the $N \times 784$ inputs to the linear read-out of an SNN, what would be the accuracy?

RESPONSE:

This is an interesting question. The main take from the mutual information calculation is that as the device continuously receives signals, its state evolves and exhibits more information about the digit classes/labels. The system's embedded memory is enabled by the short-term fading memory from the recurrent structure of the network and long-term internal memory of the individual memristive junctions. SNNs do not demonstrate explicit states other than the membrane potentials. From our results on the sequence memory task (i.e. memory enhances learning), we can infer that the accuracy will be lower using an SNN.

- I don't understand the significance of table 1. The number of samples for convergence is the same. It is not clear why there is the claim that online learning can converge faster.

RESPONSE:

Table 1 suggests that the accuracies of online methods are higher than those of batch-based methods, and using data from more readout channels increases the accuracy as well. The faster convergence of online methods was demonstrated by Farhang-Boroujeny¹ and Jaeger et al.^{2,3}.

- In the response letter, the authors mention that this is different from reservoir computing as the computational graph resembles Small World connectivity. What's the significance of small-world connectivity is solving temporal tasks?

RESPONSE:

Connectivity is just one of the differences. The other differences include internal memory (in addition to fading memory) and non-static, self-adjusting weights. The small world connectivity introduces heterogeneity into the dynamic weights, which enhances the diversity of network states and hence, readout features used for learning⁴.

- How do the authors foresee the embedded system using NWNs? Would it be possible to have this whole system with the read-out, which now includes a microcontroller, in one system?

RESPONSE:

Yes, this is certainly possible and is something we are working towards.

- How do the systems with NWNs generally predicted to scale? As the other reviewer also mentioned, how reliable is the computation from one device to another?

RESPONSE:

These systems are highly scalable as the networks are easy to synthesise. Supplementary Figure S. 7 demonstrates how different device configurations and operation can lead to varying performance, but the differences in accuracies remain relatively small. The challenging part is to ensure that a sufficient number of electrodes make electrical contact with the network.

References:

1. Farhang-Boroujeny, B. *Adaptive Filters: Theory and Applications*. (John Wiley & Sons, 2013).
2. Jaeger, H. The “echo state” approach to analysing and training recurrent neural networks – with an Erratum note. 47 (2001).
3. Jaeger, H. Adaptive Nonlinear System Identification with Echo State Networks. *NIPS* (2003).
4. Loeffler, A. *et al.* Modularity and multitasking in neuro-memristive reservoir networks. *Neuromorphic Comput. Eng.* **1**, 014003 (2021).

REVIEWER COMMENTS

Reviewer #1 (Remarks to the Author):

Some of my comments remain, as they have not been addressed.

- There should be an introductory sentence saying NWNs act like a reservoir. It will help the reader.
- The "online learning" part done on a microcontroller and the sequence learning task is a good engineering effort, but not a ground breaking idea.
- Please add a simulation without a reservoir layer to show the jump in accuracy as a result of the NWN.
- In other words, what is the fading memory of the NWN, and what is the temporal information in your task?
- Please include the small-world significance answer to the paper.

NCOMMS-23-10545A Author Responses to Reviewer Comments V3

- There should be an introductory sentence saying NWNs act like a reservoir. It will help the reader.

RESPONSE:

A sentence has been added to introduction, page 3, lines 6-10, to describe the behavior of NWNs as reservoirs:

“Previous experimental [34, 55, 56] and simulation [34, 55, 57-62] studies have demonstrated NWNs exhibit fading memory and can effectively project input signals to a higher dimensional feature space, thus enabling their use as physical reservoirs in an RC approach to machine learning.”

- The "online learning" part done on a microcontroller and the sequence learning task is a good engineering effort, but not a ground breaking idea.

RESPONSE:

The learning tasks implemented in this work (MNIST classification and sequence memory) have never been demonstrated in a NWN device before. This demonstrates the novelty of our study. The online algorithm we utilized in these tasks amplifies this achievement and opens the door to real-world applications with streaming sensor data.

- Please add a simulation without a reservoir layer to show the jump in accuracy as a result of the NWN.

RESPONSE:

The table below compares the performance of the sequence memory task without the NWN (using raw MNIST digit data) and with the NWN (using its readouts). The results show that for every single digit, performance is better with the NWN.

Digit	1	4	7	5	1	4	7	9
SSIM without NWN	0.22	0.23	0.23	0.21	0.22	0.23	0.24	0.24
SSIM with NWN	0.48	0.43	0.48	0.36	0.47	0.41	0.46	0.41

For the MNIST classification task, the mean accuracy without the NWN is 84.7% using the online RLS algorithm without any error correction. The corresponding result with the NWN is 90.5%. The table below shows the classification accuracy for each digit without and with the NWN. [Note that we achieved a higher accuracy of 93.4%, as listed in Table 1, by including active error correction in the online RLS algorithm. We have also included an additional Table 1 in Supplementary to compare results for different testing samples.] The following table compares digit-wise classification accuracies without the NWN and with the NWN. The improvements for “hard” digits like “2”, “5”, “8” are particularly noticeable.

Digit	0	1	2	3	4	5	6	7	8	9
Acc without NWN	93.3%	96.9%	79.4%	83.6%	89.8%	69.7%	92.6%	85.6%	73.4%	81.2%
Acc with NWN	94.1%	97.9%	88.6%	89.9%	94.0%	86.7%	94.7%	91.0%	80.8%	86.6%

To be continued on next page.

The difference for MNIST classification is less than that for the memory sequence task because the classification task relies less on the NWN's memory properties. Both tasks rely on the NWN's ability to extract dynamical features from the respective data. The figure below shows the input-output mapping distributions of the MNIST digit data (60,000 samples): blue is the input MNIST data and orange is the NWN readout data. Fig. 2 in the manuscript shows this mapping in more detail for 2 sample digits.

- In other words, what is the fading memory of the NWN, and what is the temporal information in your task?

RESPONSE:

The fading memory of the NWN is evident in the conductance responses to the MNIST digit voltage pulses shown in Fig. 7. An additional supplementary figure (Supplementary Figure S9 has been included to show this more clearly by zooming in and overplotting the input voltage pulses:

Supplementary Figure S9. Input voltage and network conductance for different MNIST digit inputs. Network conductance gradually decays after high intensity pixels are delivered, demonstrating the fading memory property.

The figure clearly demonstrates that when low intensity pixels are delivered to the network, the conductance gradually decays rather than immediately dropping in tandem with the voltage. This provides evidence for the presence of fading memory within the network.

The temporal information in our task is the time series of the MNIST handwritten digits (i.e. pixel intensities delivered as 1D temporal streams) which are then classified in a streaming online manner (likewise for the sequence task). Usually, the MNIST classification task is performed statically in a batch-based way. Showing that our device can perform online dynamical learning on streaming image data has important implications for real-world applications at the sensor edge (e.g. classification of streaming video data).

- Please include the small-world significance answer to the paper.

RESPONSE:

Small-worldness means the network structure is heterogenous and not completely random. The significance of this is twofold:

- (i) It is similar to biological neural networks and is suggested to be important for human brain's efficient information processing capabilities. A recent study by Suarez et al. investigated various connectomes, including that of biological brains, for reservoir computing¹. Their results and previous findings suggest that a small-world topology contributes to the echo-state property of reservoirs and therefore enhances their computational capacity².
- (ii) Unlike random networks, in which signals can propagate indiscriminately, small-world networks enable information to propagate along discrete preferred pathways (conductance pathways in NWNs), which we showed in our previous studies to improve task performance^{3,4}.

1. Misić, B. *et al.* *conn2res: A toolbox for connectome-based reservoir computing*.
<https://www.researchsquare.com/article/rs-3084265/v1> (2023) doi:10.21203/rs.3.rs-3084265/v1.
2. Kawai, Y., Park, J. & Asada, M. A small-world topology enhances the echo state property and signal propagation in reservoir computing. *Neural Netw.* **112**, 15–23 (2019).
3. Zhu, R. *et al.* Information dynamics in neuromorphic nanowire networks. *Sci. Rep.* **11**, 13047 (2021).
4. Loeffler, A. *et al.* Modularity and multitasking in neuro-memristive reservoir networks. *Neuromorphic Comput. Eng.* **1**, 014003 (2021).

REVIEWERS' COMMENTS

Reviewer #1 (Remarks to the Author):

The authors have addressed my questions. Thank you!